# Histamine Intolerance—A Kind of Pseudoallergic Reaction

**DOI:** 10.3390/biom12030454

**Published:** 2022-03-15

**Authors:** Ying Zhao, Xiaoyan Zhang, Hengxi Jin, Lu Chen, Jiang Ji, Zhongwei Zhang

**Affiliations:** 1Department of Dermatology, The Second Affiliated Hospital of Soochow University, Suzhou 215000, China; nanjjzhaoying@126.com (Y.Z.); hiohikari@163.com (X.Z.); chenlu1275@163.com (L.C.); 2Department of Dermatology and Venereology, Suzhou Medical College of Soochow University, Suzhou 215000, China; jinhengxi1995@163.com; 3Department of Biochemistry and Molecular Biology, School of Medicine, Nantong University, Nantong 226001, China

**Keywords:** histamine, histamine intolerance (HIT), diamine oxidase (DAO), differential diagnosis of HIT, treatment of HIT

## Abstract

Histamine intolerance (HIT) is a common disorder associated with impaired histamine metabolism. Notwithstanding, it is often misdiagnosed as other diseases because of its lack of specific clinical manifestations. HIT did not gain traction until the early 21st century. In this review, we will focus on the latest research and elaborate on the clinical manifestations of HIT, including its manifestations in special populations such as atopic dermatitis (AD) and chronic urticaria (CU), as well as the latest understanding of its etiology and pathogenesis. In addition, we will explore the latest treatment strategies for HIT and the treatment of specific cases.

## 1. Introduction

HIT (also known as enteral histaminosis or sensitivity to dietary histamine) is a disease that was only discovered and defined in this century [1]. It has a cumulative prevalence of 3–6% and occurs much more frequently in children [2]. Due to its nonspecific clinical manifestations (itching, flushing, edema, postprandial abdominal distension, diarrhea, abdominal pain and constipation, dizziness, headache, hypotension, tachycardia, etc.) [3], HIT was often misdiagnosed as other diseases in the past (e.g., food allergy [4,5], irritable bowel syndrome [6,7], other food intolerance, celiac disease [8], eosinophilic gastroenteritis [9], urticaria, systemic mastocytosis [10], etc.). In the past decade, HIT has gained wider social and scientific recognition, and a large number of facts about this disease have been revealed. It has been found that HIT is mainly caused by genetics [11,12,13,14,15,16,17,18,19,20], intestinal microbiome disorders [21,22], food and drugs [11,23,24,25,26,27,28,29], and other related diseases (e.g., allergic diseases, intestinal diseases, chronic infections, and mastocytosis, etc.) [16,17,19,30,31,32,33]. Based on this, we propose some detection methods that might help identify HIT, such as DAO (a kind of enzyme considered for the gastrointestinal degradation of histamine) concentration and activity [34,35], the oral histamine-challenge test [4,5], the histamine 50-skin-prick test [36], intestinal biopsy, the determination of histamine in blood [37], the detection of histamine and its metabolites in urine [38], diagnostic therapy, single-nucleotide polymorphism (SNPs) for DAO gene assessment, etc. Several symptomatic therapies have been proposed, including antihistamine therapy [24,39], a low-histamine diet [25,40,41,42,43,44,45,46,47,48], supplementation with DAO [49], treatment of severe urticaria caused by HIT [50], and other adjunctive therapies. We believe that our review may help to summarize and establish a set of up-to-date approaches to the etiology, diagnosis, and treatment of HIT.

## 2. Histamine

### 2.1. Cellular Origin

Histamine (2-(4-imidazolyl)-ethylamine) is a biogenic amine that is widely present in the human body and can have a wide range of physiological and pathological effects, such as allergy and inflammation, by binding to its specific receptors (H1R, H2R, H3R, H4R) [51,52,53]. It is generally believed that abnormal histamine accumulation in the body is caused by a decrease in the ability of enzymes to degrade histamine. Histamine can be synthesized and released by a variety of cells, including but not limited to mast cells, basophils, chromaffin cells, platelets, dendritic cells, T cells, histaminergic neurons, etc. Mast cells and basophils are the primary cellular sources of histamine [53]. In these cells, L-histidine is decarboxylated in the Golgi apparatus by histidine decarboxylase (HDC) to form histamine, which is further stored in cytoplasmic granules. Mast cells and basophils are sensitized and degranulated, and then histamine is released. Mast cells have been shown to be present in skin, intestines, etc., which means histamine is also abundant in these areas [54].

### 2.2. Foods Rich in Histamine

Histamine has many other sources besides cells, such as foods [55,56]. Common histamine-rich foods include fish and seafood, aged or fermented foods (such as bacon, cheese, pickles, etc.), and some vegetables (such as spinach, eggplant, tomato, etc.). The formation of biogenic amines in food requires the availability of free amino acids, the presence of decarboxylase-positive microorganisms, and conditions allowing bacterial growth and decarboxylase activity. Free amino acids can either be present in the food itself or released by proteolysis during processing or storage [57]. Histidine is produced by autolytic or bacterial processes. Therefore, high concentrations of histamine are mainly present in microbial fermentation products. Foods that are rich in histamine are listed in the table below [57,58,59,60,61,62] (Table 1).

### 2.3. Histamine Metabolic Pathway

Histamine is metabolized through two main pathways: the cyclopentyl action of histamine-N-methyltransferase (HNMT) and the oxidative deamination of DAO [38,40,63,64]. Under the act of HNMT, a small part of histamine is converted to N-methylhistamine, most of which is then converted to N-methylimidazolium acetic acid with the participation of monoamine oxidase. Most of the histamine is converted to imidazole acetic acid with the help of DAO. About two-thirds of the imidazole acetic acid binds to ribose and is excreted in the urine, while the remainder is excreted in its original form. About 2–3% of histamine is excreted in the urine in its original form [32]. HNMT is a kind of cell-soluble lipoprotein, which is mainly responsible for the transformation of intracellular histamine. DAO, on the other hand, is a secreted protein accountable for the removal of extracellular histamine. Notably, evidence suggests that alcohol (especially red wine) and acetaldehyde can increase endogenous histamine release and competitively interact with acetaldehyde dehydrogenase to reduce the metabolic rate of histamine, which contributes significantly to the increase in histamine levels in vivo [11,24,65] (Figure 1).

## 3. Histamine Intolerance

HIT is a non-immune reaction, which we call a pseudoallergic reaction in the title. It is food intolerance caused by histamine accumulation and ingestion [1]. Various reasons lead to a decrease in the level or activity of histamine-metabolizing enzymes and an increase in exogenous histamine, resulting in an increase in histamine and a series of discomforts in the body. This phrase needs to be distinguished from histamine poisoning. Histamine poisoning is also called a scombroid syndrome, scombroidosis. The word originated from the name of the Scombridae family. After eating this fish, poisoning can often be seen. Histamine poisoning is considered to be one of the most common poisonings in the world caused by eating fish [66]. The manifestations of histamine poisoning may include skin rash, abdominal pain, vomiting, diarrhea, and shortness of breath, and may also lead to fatal consequences [67].

### 3.1. Causes of HIT

#### 3.1.1. Genetically Induced HIT

The related enzymes that degrade histamine are DAO and HNMT. The DAO gene is located at 7q35 and is about 10 kb in length, and the HNMT gene is located at 2q22.

HIT is mainly caused by SNPs in the DAO gene. The expression of this gene leads to changes in protein production, and enzyme activity is lower than normal. The most important SNPs in the DAO gene are rs10156191, rs1049742, rs2268999, and rs104979. SNPs in the promoter region (rs2052129) have also been reported to reduce DAO gene expression [11]. Various SNPs of the DAO gene are related to inflammatory and neoplastic gastrointestinal (GI) diseases, such as food allergy [12], gluten-sensitive enteropathy, Crohn’s disease, ulcerative colitis, and colon adenoma [13,14,15]. A number of studies have investigated the relationship between SNPs in the HNMT gene and allergic diseases in children [16,17,18,19,20]. HNMT SNPs seem to be related to the more serious course of AD and allergic rhinitis. Therefore, HIT seems to be mainly caused by impaired DAO activity caused by GI diseases or by inhibiting DAO. The high degree of inter-individual variability of DAO expression in the intestine and the association of SNPs in the DAO gene with GI diseases provide evidence for the genetic susceptibility of this subgroup of histamine-intolerant patients.

#### 3.1.2. HIT Associated with Dysbacteriosis

In addition to the impaired degradation of oral histamine due to a lack of DAO, disturbances in the intestinal flora may also lead to increased histamine levels. Some bacteria also seem to synthesize and secrete histamine [68,69]. Martin Hrubisko et al. elegantly summarized and introduced histamine-producing bacteria in detail in their review [2]. Schink et al. conducted a 64-person study [21], including 8 HIT patients, 25 with food hypersensitivity, 21 with food allergy, and 10 healthy controls. No significant differences in stool histamine levels were observed, but the stool zonulin levels were elevated in HIT patients. The analysis of microbial flora showed that the level of *Proteus* in the HIT group increased, and the α diversity decreased significantly. The abundance of *bifidobacteria* in HC was significantly higher than that in other research groups, while the abundance in the HIT group was the lowest. In patients with HIT, the abundance of the genera *Butyricimonas* and *Hespellia* was significantly decreased, and the abundance of *Roseburia* was significantly increased. Therefore, the changes in *Proteus* and *Bifidobacteriaceae*, the decrease in α diversity, and the increase in fecal zonrin levels suggest that patients with HIT have ecological disorders and intestinal-barrier dysfunction.

The presence of bacteria, yeasts, and molds that form histamine in the GI tract may increase the sensitivity of some people to histamine ingestion. Frei et al. studied the effects of histamine-producing strains of *Lactobacillus rhamnosus* on mice and found that both host- and microbiota-derived histamine significantly altered the innate immune response to microbes through H2 receptors [22].

#### 3.1.3. HIT Caused by Related Diseases

HIT may be a comorbidity in most cases. Studies on DAO/HNMT deficiency or increased serum histamine levels are associated with allergic diseases (such as AD, allergic rhinitis, or asthma), and the severity of bronchial hyper-responsiveness supports this view [16,17,19]. In addition, DAO deficiency is present in intestinal diseases, such as colon adenoma, carbohydrate malabsorption, food allergies, and anergy celiac disease (CD) [30,31,32,33]. Chronic infections, mastocytosis, etc., can also increase the histamine in the body, resulting in symptoms of HIT. People suffering from the above diseases are prone to DAO deficiency. Due to the increase in serum histamine concentration, the frequency or severity of histamine-mediated symptoms may increase, leading to the coexistence of HIT. DAO deficiency caused by these diseases may be another cause of HIT.

#### 3.1.4. Drug-Induced HIT

The causes of histamine overdose are related to the widely used phlegm-reducing agents (ambroxol, acetylcysteine), antiemetics (metoclopramide), antiarrhythmic drugs (verapamil, pramalin), anti-arrhythmic drugs, hypertension drugs (dihydro hydrazine), alcohol, antidepressants (amitriptyline), chloroquine, and clavulanic acid inhibiting the activity of DAO [23,24]. After this temporary inhibition is released, HIT will be relieved.

#### 3.1.5. Food-Induced HIT

Sauerkraut, fish, smoked meat products, and some kinds of cheeses contain large amounts of histamine, which may contribute to HIT symptoms due to a slight excess of histamine [11,25,26,27]. To diagnose HIT, patients who consume histamine foods must be accompanied by insufficient DAO. Otherwise, it should be diagnosed as histamine poisoning. Some foods may trigger the release of histamine; for example, papaya, strawberries, citrus fruits, some nuts, egg whites, chocolate, fish, and pork. Foods that are rich in other biogenic amines may cause competitive DAO inhibition, exceeding the usually tolerable amount of histamine, which may cause HIT [11,24]. Fermented sausage, fish and sauerkraut are such foods. These foods contain both histamine and more cadaverine or putrescine. DAO can degrade putrescine, histamine and cadaverine. The cadaverine or putrescine in these foods will compete with histamine for binding sites on the DAO, thereby inhibiting the degradation of histamine by DAO [70]. Putrescine can promote the release of histamine from the intestinal mucosa. Foods that are rich in putrescine include fermented sausage, cheese, fish sauce, green peppers, citrus fruits, wheat germ and bean sprouts [28,29] (Table 2). Some kinds of foods can cause acquired DAO deficiency through several mechanisms at the same time. For example, fish is a food that is rich in histamine, promotes histamine release, and inhibits DAO.

In summary, HIT is a food intolerance caused by multiple factors. The etiology of HIT is mainly related to genetic polymorphisms of DAO or HNMT, diseases associated with DAO or HNMT deficiency, dysbacteriosis, drugs that inhibit DAO activity, foods that are rich in histamine, foods that inhibit DAO, and foods that promote histamine release (Figure 2).

### 3.2. Pathophysiological Mechanism of HIT

Due to the above causes of histamine accumulation in the body, exogenous or endogenous histamine is involved in various clinical responses of systemic organs by binding to four types of receptors (H1–H4 subtypes) [71]. H1-subtype receptors are located on the surface of cells such as the vascular endothelium and airway smooth muscle. Once bound, they are involved in allergic reactions and vasodilation. H2-subtype receptors are mainly located on various cells such as the mucosa, epithelium, immune cells and smooth muscle, and are involved in gastric-acid secretion, smooth-muscle-cell relaxation, and immune-cell differentiation. H3 receptors are expressed in the nervous system as presynaptic autoreceptors and play an important role in the regulation of neural function cognition and neuronal histamine turnover. H4 receptors are expressed on a variety of immune cells, including keratinocytes, Langerhans cells, neutrophils, lymphocytes and dendritic cells, and are involved in immune regulation, including immune-cell chemotaxis, immune responses and inflammation. Too much histamine causes disorder in the above reactions, leading to various clinical manifestations in patients with HIT [56].

## 4. Clinical Manifestations of HIT

### 4.1. Skin

The skin contains histamine receptors [72], which can cause vasodilation and other physiological reactions. As a result, histamine can affect the skin in many ways. Many types of allergic skin diseases are caused by histamine. The skin symptoms of HIT can be classic and diverse. Schnedl et al. have reviewed the symptoms of 133 HIT patients [3]. The symptoms of these patients included pruritus, flush, edema, erythema, eczema, and so on, with pruritus being the most common. There were 48 patients with pruritus. Some of them had multiple symptoms. The skin symptoms were mostly pruritus, flush, and edema. Some patients developed a wheal-like rash that looked like a symptom of urticaria.

### 4.2. GI Tract

The GI tract contains almost all of the histamine receptors. As a result, patients with HIT are often accompanied by many GI symptoms, including diffuse stomachache, GI colic, flatulence, diarrhea, and so on [3]. Elevated histamine concentrations and decreased DAO activity have been found in a variety of inflammatory and neoplastic bowel diseases [73]. Decreased HNMT and impaired total-histamine degradation were detected in colonic mucus from patients with food allergies. A questionnaire conducted by Wolfgang J. Schnedl et al., based on 133 patients with HIT, also found that 92% of patients had abdominal distension, 73% had postprandial abdominal distension, 71% had diarrhea, 68% had abdominal pain, and 55% had constipation. Additionally, there are common symptoms such as nausea and vomiting [3].

### 4.3. Respiratory System

When ingesting foods that are rich in histamine, patients with HIT may have respiratory symptoms such as a runny nose, nasal obstruction and nasal mucosal congestion. Some severe patients even have asthma symptoms [3].

### 4.4. Other Manifestations

The four types of histamine, H1, H2, H3 and H4, are widely distributed in the human body [74,75]. The latest study by Schnedl et al. found that HIT in the nervous system mainly manifested as dizziness, headache, migraine, and rhinorrhea. Many patients with headache symptoms are relieved after taking antihistamines [76]. In gynecology, it is mainly manifested as dysmenorrhea. In the cardiovascular system, the main manifestations are tachycardia and hypotension [3] (Figure 3).

### 4.5. Special Cases

#### 4.5.1. Atopic Dermatitis (AD)

HIT is reported to occur more frequently in some AD patients because of decreased DAO activity and increased histamine concentration [77,78]. The symptoms of AD and HIT are all relieved by a low-histamine diet [46].

#### 4.5.2. Chronic Urticaria (CU)

Histamine is the primary mediator of wheal and angioedema in CU, so HIT is also specific in patients with CU. HIT has also been considered as a cause of or contributing factor to CU. According to relevant reports, symptoms of some CU patients can be erased after a low-histamine diet [47,79]. Meanwhile, the symptoms of CU can be relieved by complementary therapy with DAO [80]. Siebenhaar et al. showed that oral histamine-rich foods did not cause symptoms of HIT in patients with CU who had no previous history of HIT [81]. The result may suggest that CU is not a cause of HIT.

## 5. Differential Diagnostic Exclusion of Other Diseases

### 5.1. Food Allergy

Food allergy can also be manifested as abdominal pain, diarrhea, itching, urticaria, angioedema and so on. It can also occur after eating. However, food allergy is caused by an allergic reaction, while HIT is not. In laboratory examination, patients with food allergy may have elevated IgE, while patients with HIT have normal IgE. In addition, the gold standard of food allergy is the oral food-challenge test, while the specificity of the oral histamine-challenge test is not high [4,5]. Both HIT and food allergy can be relieved with antihistamines.

### 5.2. Irritable Bowel Syndrome (IBS)

Clinicians often classify many clinical symptoms of food intolerance as IBS. At present, the diagnostic standard of IBS is the Rome IV criteria [82]. However, the clinical symptoms of IBS are imprecise, lack specificity, and have no sound pathophysiological basis. Therefore, IBS seems more appropriate to be defined as a comprehensive term for GI-related disorders rather than as a separate disorder [6]. It is worth mentioning that food antigens (including but not limited to histamine) are responsible for approximately 80% of GI symptoms in patients with IBS. Thus, there may be overlapping relationships between HIT and IBS [7].

### 5.3. Other Food Intolerances and Celiac Disease (CD)

The clinical manifestations of HIT are often very similar to intolerance diseases such as sugar (lactose and fructose), protein (gluten), sulfite, or other biogenic amines (such as tyramine), which are often misdiagnosed or missed. The onset of HIT is often associated with the intake of high histamine in foods (such as fish and aged or fermented foods) or beverages (such as red wine) [55,56]. Moreover, patients with HIT tend to have a decrease in their serum-DAO values. Patients with lactose and fructose intolerance showed a positive H_2_ breath test. Therefore, an identification can be made based on the type of food consumed and the serum DAO level. Interestingly, HIT patients also seem to present with disrupted gut flora [21]. This may be related to the destruction of the intestinal mucosa and the high expression of DAO in the intestine.

CD is mainly caused by the degranulation of mast cells and the release of histamine and other inflammatory mediators after the ingestion of gluten [8]. CD is confirmed by serological examination (antibodies against tissue transglutaminase with IgA ELISA) and duodenal biopsy. A large number of patients with refractory CD complicated with HIT have been found. A study based on 20 patients with refractory CD found 11 patients with HIT. In addition, some foods containing gluten, such as bread, noodles, oatmeal, and beer, also contain high levels of histamine, which means refractory CD may be combined with HIT.

### 5.4. Helicobacter Pylori (HP) Infection

HP can lead to changes in the stomach environment that affect the absorption of nutrients, resulting in some GI symptoms. HP infection can often be confirmed by a C_13_ breath test, serological antibody test, and GI endoscopy combined with a GI mucosal histological evaluation and can be cured by the specific antibiotic. Conversely, these treatments have no effect on HIT.

### 5.5. Eosinophilic Gastroenteritis (EGE)

The GI symptoms of EGE are somewhat similar to those of HIT [9]. The detailed mechanism of EGE remains unclear, but it is mainly caused by eosinophils infiltrating into the GI mucosa. In a disordered GI tract, mast cells and eosinophils depend on each other, thus activating and causing functional GI disorders. ECE can be alleviated by the administration of corticosteroids. A clinical diagnosis can also be made by measuring the levels of serum trypsin and DAO.

### 5.6. Urticaria

When patients with HIT develop urticaria-like rashes, the HIT should be differentiated from other types of urticaria, such as spontaneous urticaria and artificial urticaria. Patients with HIT often have a history of eating foods that are rich in histamine. In laboratory examination, the DAO concentration is decreased in HIT.

### 5.7. Systemic Mastocytosis

Histamine is released by mast cells, so the clinical manifestations of histamine overdose caused by systemic mastocytosis are similar to those caused by HIT [10]. However, HIT is associated with a decrease in DAO and changes in serum trypsin, which are absent in systemic mastocytosis (Figure 4).

## 6. Methods for Detecting HIT

### 6.1. History and Clinical Manifestations

HIT can be inferred or suspected by the aforementioned clinical manifestations. However, it is necessary to consider the patient’s history due to the low specificity and complexity of the HIT clinical manifestations. Moreover, it is also important to consider whether the clinical manifestations are related to eating foods that are rich in histamine or taking drugs that inhibit DAO activity.

### 6.2. DAO Concentration and Activity

DAO plays the role of degrading exogenous histamine. As a result, its expression is very important for the diagnosis of HIT. According to relevant tests, the DAO concentration and activity in patients with HIT were significantly lower than those in normal subjects [34,35]. Therefore, DAO concentration and activity can be used as a diagnostic method for HIT. The method to detect them is ELISA.

### 6.3. Oral Histamine-Challenge Test

This test involves an oral 75 mg histamine solution to observe the clinical reaction. The trial may cause dangerous clinical symptoms, so it must be carried out under the supervision of a doctor and in preparation for rescue in the hospital. According to the studies [4,5], the diagnostic significance of this test is not like the oral food-challenge test for food allergies. It causes a more random set of clinical symptoms in which the same patient may present with different reactions or may not. Sometimes healthy individuals will develop HIT symptoms. For the above reasons, the oral histamine-challenge test is not recommended as a diagnosis for HIT because of its high risk and low diagnostic value.

### 6.4. Histamine 50-Skin-Prick Test

With 0.9% of physiologic saline solution serving as a negative control, the intact skin is pipetted with 1% (10 mg/mL) histamine solution, and the skin is punctured with a lancet. A wheal ≥ 3 mm within 50 min is considered positive. The experiment of Lukas suggests that the sensitivity of the histamine 50-skin-prick test is 79% (95% confidence intervals: 68.5–87.3%), and the specificity is 81.3% (95% confidence intervals 70.7–89.4%) [36]. It can be used as an auxiliary diagnostic method for HIT.

### 6.5. Intestinal Biopsy

As mentioned above, HIT is related to the lack of intestinal DAO. As a result, intestinal biopsy through a gastroscopy to detect the concentration of intestinal DAO can be a highly sensitive and specific method. According to the research of [21], HIT is also related to intestinal dysbacteriosis. Intestinal biopsy for the detection and culture of intestinal flora can also assist in the diagnosis of HIT. When patients have functional gastrointestinal symptoms, gastroscopy can also be used as a valuable adjunct to examination [83]. However, this kind of method is an invasive examination with associated risks, and the pros and cons need to be weighed. Good communication should be maintained between doctors and patients before intestinal biopsy.

### 6.6. Diagnostic Therapy

It has been proven that a low-histamine diet and DAO supplement could be the main treatment for HIT, since they can effectively relieve the symptoms of HIT [25,42,49]. The low-histamine diet is a lifestyle-altering treatment with no need for medication, while the DAO supplement is also convenient for patients. As a result, the therapy is noninvasive and easy for patients. If patients have the corresponding clinical manifestations of HIT, which is hard to diagnose, then doctors can treat them with a low-histamine diet or supplementation with DAO and observe whether the HIT symptoms relieve. If their symptoms significantly improve after 4–8 weeks of diagnostic treatment, it would be highly suspected as HIT. This can be used as one of the diagnostic criteria of HIT.

### 6.7. Determination of Histamine and Its Metabolites in Urine

According to recent studies, the detection of histamine and histamine metabolites in urine can also be used as a diagnostic method for HIT [38]. Doctors can test the histamine and 1-methylhistamine in human urine, determined by UHPLC-FL. Compared with the determination of DAO concentration, this method is more convenient in terms of time consumption, sample collection and equipment, and as a result, more acceptable to patients. This method could be a potential routine diagnostic method in clinical practice.

### 6.8. Determination of Histamine in Blood

There is a lack of histamine degradation due to the lack of DAO in patients with HIT. Therefore, the elevation of histamine in the blood can be used as one aspect of the diagnosis of HIT. Current methods for detecting histamine include colorimetry, fluorescence, ELISA and chromatography, which are expensive and inconvenient [37]. Histamine needs better detection methods.

### 6.9. SNPs of DAO Gene Assessment

HIT is related to the deficiency of histamine degradation by DAO in the intestine. Therefore, the SNPs of these two can be used to estimate the susceptibility of HIT. SNPs are assessed by sampling blood or oral mucosa.

### 6.10. To Rule Out Other Diseases with Similar Symptoms

All of the above methods of diagnosing HIT need to exclude symptomatic diseases and other concomitant GI disorders, as well as related diseases mentioned in the previous differential diagnosis (Table 3).

## 7. Treatments of HIT

### 7.1. Antihistamine

H1- and H2-receptor antagonists can be taken after taking a large amount of histamine, but they cannot be used for long-term treatment of HIT [84]. Most antihistamines do not affect DAO activity, but there are reports that cimetidine and dihydralazine can inhibit DAO activity, while diphenhydramine can increase DAO activity [24]. There are also reports in the literature that the effect of a strict no-histamine diet plus antihistamine drugs is not enhanced [39]. The choice of the therapeutic dose and the production and type of antihistamines (H1/H2) is determined by the clinician after taking into account the symptoms (GI tract, nerves, skin), but in terms of effectiveness and safety, priority should be given to the use of second-generation or third-generation H1-receptor antagonist. H2-receptor blockers can be used for patients with GI symptoms.

### 7.2. Limit Histamine

Based on the diagnosis of HIT, a low-histamine diet was developed to treat individual symptoms by reducing the intake of histamine-containing foods [1]. This aims to exclude foods with high histamine content, and the histamine content in foods varies greatly depending on the storage time and processing method [40]. In fact, not only for the high content of histamine, some foods that promote the release of histamine and inhibit DAO should also be avoided. At present, many studies have made specific recommendations about foods to avoid for people with HIT [25,41,42,43,44,45,46,47,48]. However, after comparison, it has been found that the excluded foods are different in most suggestions, depending on the diet considered. However, fermented foods are consistently excluded. The exclusion of 32% of foods can be explained by the high histamine content. The presence of putrescine may prevent histamine from degrading by the DAO at the gut level, which may partly explain why certain foods (i.e., citrus fruits and bananas) are also frequently present in low-histamine diets. The last food to avoid are foods that promote the release of histamine, such as citrus fruits, papaya, strawberries, egg whites, chocolate, nuts, fish, pork, cheese, fermented sausage, green peppers, wheat germ and bean sprouts [85].

### 7.3. Supplement of DAO

Schnedl et al. conducted a study on 28 HIT patients and instructed them to take DAO capsules before meals for four consecutive weeks. During the oral DAO process, GI, respiratory, cardiovascular and skin symptoms were significantly improved [49]. The European Food Safety Agency (EFSA) has approved a pig-kidney extract containing 0.3 mg of DAO as a new food. Since 2002, it has been sold on the market as a nutritional supplement, and since 2013, it has been sold on the market as a food for special medical purposes. According to the EFSA, the maximum daily dose of the exogenously ingested enzyme is 3 × 0.3 mg, which is equivalent to 0.9 mg of DAO.

### 7.4. Treatment of Severe Urticaria Caused by HIT

Severe HIT may have life-threatening reactions, such as hypotension, bronchospasm, shock, etc., presenting a state of anaphylactic shock. 

Generally, this serious reaction is related to high-dose, high-histamine substances. In this case, first aid is needed [50]. First, stop ingesting foods that cause HIT, quickly establish venous access, and monitor vital signs. If there is a drop in blood pressure and a faster heart rate, immediately replenish blood volume. Then, inject 0.1% epinephrine 0.3–0.5 mL into the muscle and inject dexamethasone or hydrocortisone into the vein. If the blood pressure still does not rise after the above measures, vasoactive drugs (such as dopamine, etc.) can be used. Antihistamine drugs can be used to counteract the accumulation of histamine in the body (such as promethazine). If breathing difficulties worsen, perform first-aid measures such as tracheal intubation or tracheotomy, cardiopulmonary resuscitation (Figure 5).

### 7.5. Adjuvant Treatment of HIT

Taking vitamin C and vitamin B6 can increase the activity of DAO, which is conducive to the degradation of histamine. Mast-cell stabilizers and pancreatin have a positive effect on HIT, especially for patients with GI symptoms [25,40,46,86]. According to the aforementioned effect of flora disorders in patients with HIT, the addition of probiotics may lead to the regulation of the microbial community, thereby reducing the production of the microbial enzyme L-HDC, and more clinical trials are needed to verify the efficacy.

## 8. Conclusions

HIT is a kind of common comprehensive disease in the clinical setting, which is caused by heredity, dysbacteriosis, related diseases, drugs and food. The manifestations of HIT lack specificity. They can occur in the skin, GI tract, the nervous system, and so on. In particular groups, HIT can have a relatively different effect. Therefore, when a doctor encounters a patient with these clinical symptoms, they should consider the possibility of HIT and distinguish it from diseases with similar symptoms. The current detection methods are complex, sometimes expensive, and time consuming. There is a need to find a more convenient and rapid detection method. The existing treatments include a low-histamine diet, antihistamine therapy, DAO supplementation, and some adjuvant therapies. In the case of serious patients, the doctor should treat them in a timely and effective manner according to the methods described above in order to avoid serious consequences caused by improper or untimely treatment. Further studies are needed on the etiology, diagnosis, and treatment of HIT. I believe that we can learn more about HIT in the future.

## Figures and Tables

**Figure 1 biomolecules-12-00454-f001:**
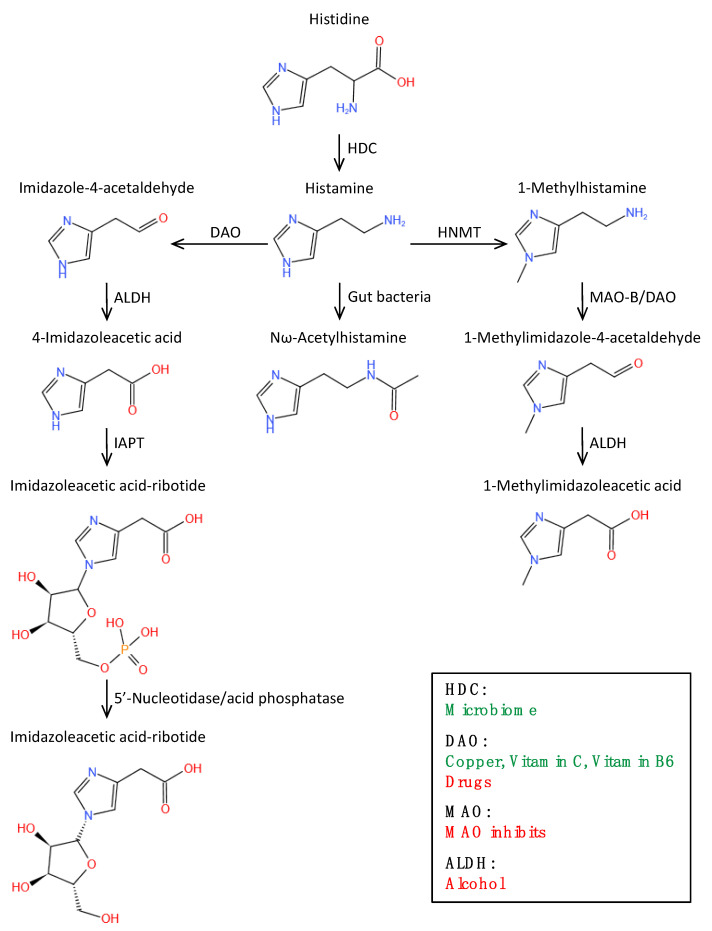
Histamine metabolic pathway. Abbreviations: HDC: histidine decarboxylase; DAO: diamine oxidase; HNMT: histamine-N-methyl transferase; ALDH: aldehyde dehydrogenase; MAO: monoamine oxidase; IAPT: imidazole acetic acid phosphoribosyltransferase. Green is the factors to enhance the endogenous ability of enzyme reaction. Red is the factors that directly/indirectly inhibits the enzyme reaction.

**Figure 2 biomolecules-12-00454-f002:**
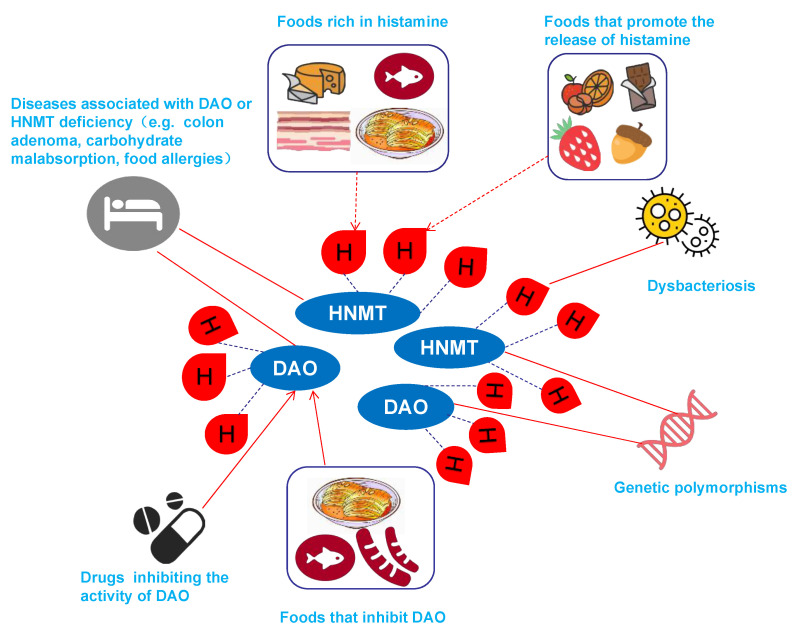
The causes of HIT. The red drop with H represents the histamine. The etiology of HIT is mainly related to genetic polymorphisms of DAO or HNMT, diseases associated with DAO or HNMT deficiency, dysbacteriosis, drugs that inhibit DAO activity, foods that are rich in histamine, foods that inhibit DAO, and foods that promote histamine release.

**Figure 3 biomolecules-12-00454-f003:**
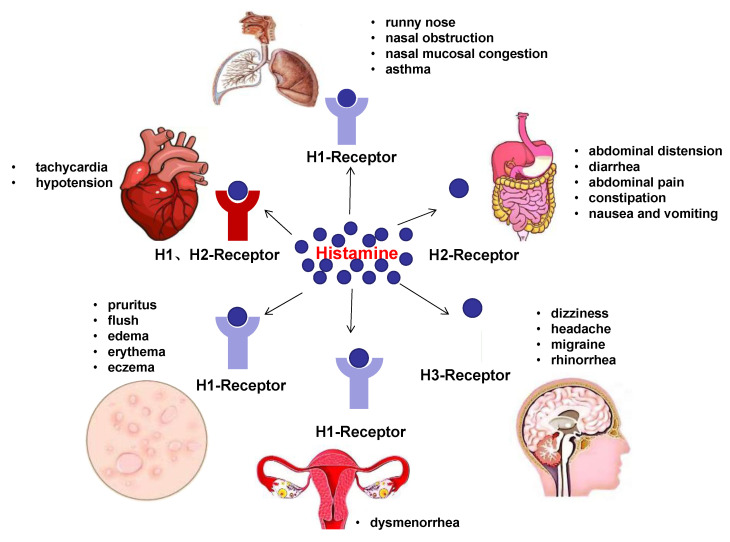
Clinical manifestations of HIT. Histamine acts on the H1, H2 and H3 receptors of the nervous system, respiratory system, cardiovascular system, skin, digestive system, and reproductive system, producing a series of clinical responses.

**Figure 4 biomolecules-12-00454-f004:**
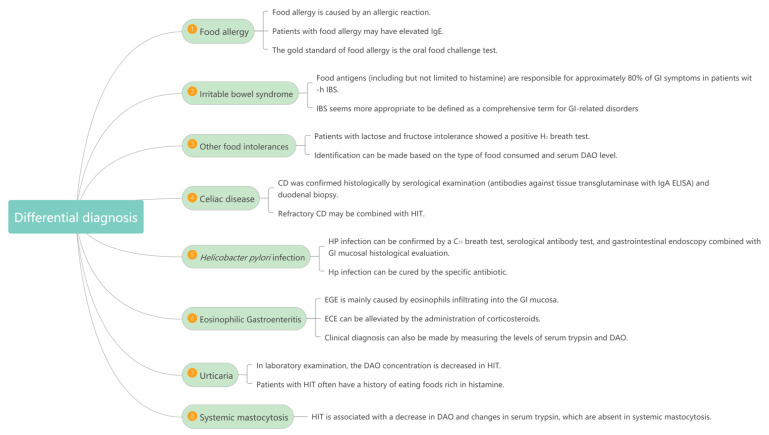
Differential diagnosis of HIT. HIT is clinically easily confused with food allergy, IBS, other food intolerances, CD, HP infection, EGE, urticaria and systemic mastocytosis. It can be distinguished based on medical history, clinical manifestations, examinations and treatments.

**Figure 5 biomolecules-12-00454-f005:**
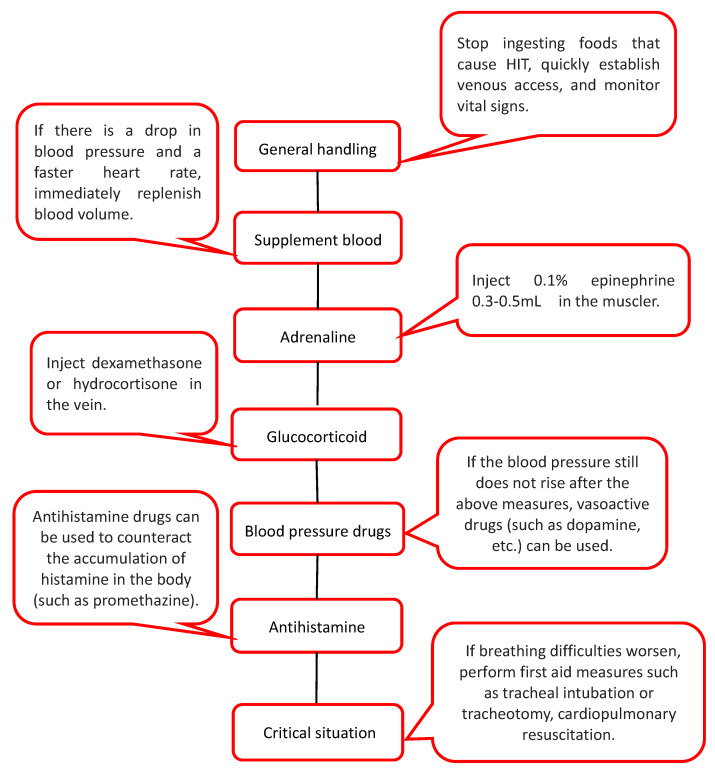
Treatment of severe urticaria caused by HIT. When severe urticaria occurs in HIT patients, the condition is critical. A timely treatment process is required, and the specific process is as above.

**Table 1 biomolecules-12-00454-t001:** Foods that are rich in histamine.

Food Categories	Histamine		Recommended Upper Limit for Histamine	
	mg/kg	mg/L	mg/kg	mg/L
Fish (frozen/smoked or salted/canned)			200	
Mackerel	1–20/1–1788/ND–210			
Herring	1–4/5–121/1–479			
Sardine	ND/14–150/3–2000			
Tuna	ND/ND/1–402			
Cheese			No recommendation	
Gouda	10–900			
Camembert	0–1000			
Cheddar	0–2100			
Emmental	5–2500			
Swiss	4–2500			
Parmesan	10–581			
Meat			No recommendation	
Fermented sausage	ND–650			
Salami	1–654			
Fermented ham	38–271			
Vegetables				
Sauerkraut	0–229		10	
Spinach	9–70			
Eggplant	4–101			
Tomato	ND–17			
Ketchup	ND–22			
Avocado	ND–23			
Red wine vinegar	4			
Alcohol				
White wine		ND–10		2
Red wine		ND–30		2
Top-fermented beer		ND–14		
Bottom-fermented beer		ND–17		
Champagne		670		

ND: not detected.

**Table 2 biomolecules-12-00454-t002:** Foods that cause HIT.

**Foods Rich in Histamine**	Fish, sauerkraut, smoked meat products and cheeses
**Foods that Promote the Release of Histamine**	Citrus fruits, papaya, strawberries, egg whites, chocolate, nuts, fish, pork, cheese, fermented sausage, green peppers, wheat germ and bean sprouts
**Foods that Competitively Inhibit DAO**	Fish, fermented sausage, and sauerkraut

**Table 3 biomolecules-12-00454-t003:** Summary of diagnostic approaches to HIT.

**Anamnesis**
Presenting ≥ 2 symptoms of histamine intolerance (details in Section 4)
Manifestation of symptoms in less than 4 h after food intake
Exclude other diseases (details in Section 6)
**Diagnostic Therapy (4–8 weeks)**
Symptoms improved after dismissing drugs interfering with histamine metabolism and distribution (details in Section 3.1.4)
Symptoms improved after using H_1_/H_2_ antihistamines (preferably short-term use, details in Section 7.1)
Symptoms improved after low-histamine diet (details in Section 7.2)
Symptoms improved after DAO supplementation (details in Section 7.3)
**Additional Tests**
Oral histamine-challenge (provocation) test
Determination of DAO concentration and activity in plasma or intestinal biopsy
Histamine 50-skin-prick test
Determination of histamine and its metabolites in urine or stool
Determination of histamine in blood
Single-nucleotide polymorphism (SNPs) of DAO/HNMT gene assessment

## Data Availability

No new data were created or analyzed in this study. Data sharing is not applicable to this manuscript.

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
