# Peer review of "Histamine Intolerance—A Kind of Pseudoallergic Reaction"

_biomolecules, 2022, doi:10.3390/biom12030454_

Round 1

Reviewer 1 Report

General comments: The authors reviewed histamine intolerance and its causes, clinical manifestations, diagnosis, and treatments. Some sections of the review are too general (e.g., Methods for detecting HIT). The authors should provide more details and elaborate on those parts. Additional comments and suggestions can be found in my specific comments.

Specific comments

Line 19: What is the prevalence of HIT?

Line 31: Define DAO.

Line 46: How can foods and drinks degrade histamine?

Table 1: Why does the histamine content in each fish has several different ranges? If these values were from different references, please clearly indicate it in the footnote.

Line 70: Consider providing a figure to illustrate the two main pathways.

Line 120: What is the difference between food hypersensitivity and food allergy?

Lines 125-132: Italicize genus and species names.

Line 134: Confusing sentence.

Line 138: What does AD stand for?

Line 154: A space is missing.

Line 166: Please elaborate how certain foods can inhibit DAO.

Line 194: Awkward sentence.

Figure 1: “nasal mucosal” is not a symptom.

Line 234: I don’t think this title reflects the contents of this section.

Line 266: ELISA is not a histological analysis.

Lines 321-323: Do not use first person.

Lines 327-329: This paragraph belongs to a separate section.

Lines 331-332: Awkward sentence.

Lines 343-345: These are all routine laboratory techniques that do not sound expensive or inconvenient.

Lines 373-374: Grammatical error.

Line 397: Change “ml” to “mL” throughout the manuscript.

Author Response

Dear Prof. Naomi Chang (Editor), and Reviewers:

Thank you for your letter and for the reviewers’ comments concerning our manuscript entitled “Histamine intolerance—a kind of pseudoallergic reaction” (Manuscript ID: biomolecules-1607389). The comments are all valuable and were very helpful for revising and improving our manuscript. We have studied the comments carefully and hope that the corrections meet with your approval. The revised portions of the paper are highlighted in yellow. The main corrections in the paper and the responses to the reviewers’ comments are as follows:

To Reviewer 1

Comments to the Author
General comments: The authors reviewed histamine intolerance and its causes, clinical manifestations, diagnosis, and treatments. Some sections of the review are too general (e.g., Methods for detecting HIT). The authors should provide more details and elaborate on those parts. Additional comments and suggestions can be found in my specific comments.

On behalf of my co-authors, we thank you very much for giving us an opportunity to revise our manuscript, we appreciate reviewer 1 for the positive and constructive comments and suggestions on our manuscript.

We have carefully examined reviewer 1s’ comments and have added writing to the general parts of the diagnostic methods to enrich the content as much as possible. However, due to space problems, we considered the blood histamine and the gene test as the parts we wanted to skim over, so the content didn't go into detail. We hope the corrections will meet with your approval.

  1. Line 19: What is the prevalence of HIT?

The prevalence of HIT has been added in lines 21-22.

  1. Line 31: Define DAO.

The definition of DAO has been added in lines 33-34.

  1. Line 46: How can foods and drinks degrade histamine?

We thank reviewer 1 for pointing out our mistake. We have changed “a decrease in the ability of specific foods, drugs, drinks, or enzymes to degrade histamine” to “a decrease in the ability of enzymes to degrade histamine”. (line 49)

Drinks and foods can produce histamine but not degrade it.

  1. Table 1: Why does the histamine content in each fish has several different ranges? If these values were from different references, please clearly indicate it in the footnote.

We thank Reviewer 1 for raising this point. The differences are because these three different ranges represent the histamine content contained in three different cooking ways of fish, as we marked after the fish in the table, they are frozen, smoked or salted, and canned.

  1. Line 70: Consider providing a figure to illustrate the two main pathways.

We thank Reviewer 1 for pointing out the inadequacies of our expression. We have added a figure (Figure 1, line 87) to illustrate the two main metabolic pathways.

  1. Line 120: What is the difference between food hypersensitivity and food allergy?

We apologize for our lack of clarity. Food hypersensitivity (FH) and food allergy (FA) are from the paper:

[21] Schink, M, et al. Microbial patterns in patients with histamine intolerance. J Physiol Pharmacol 2018, 69, doi:10.26402/jpp.2018.4.09.

It is classified as such in the original text, participants with gastrointestinal (diarrhea, nausea, vomiting, abdominal pain) and extra-intestinal symptoms (allergic rhinitis, oral allergy syndrome, headache, fatigue, skin changes, asthmatic symptoms) briefly after food ingestion and positive serological food-specific IgE antibodies and significantly elevated total IgE (361.2 kUA/L; P > 0.001) were classified as food allergy patients (FA group). The remaining participants without IgE antibodies and without validated histamine intolerance, but clinical symptoms, including abdominal pain, diarrhea, nausea, headache, skin changes, or allergic rhinitis, were classified as food hypersensitive patients (FH group).

  1. Lines 125-132: Italicize genus and species names.

We thank reviewer 1 for pointing out our oversight. We have italicized the genus and species names in lines 132-137.

  1. Line 134: Confusing sentence.

We are very sorry for this confusing sentence. We have revised this sentence to make it more clearly.

The original meaning was "both host- and microbiota-derived histamine significantly alter the innate immune response to microbes through H2 receptors. " (lines 142-144)

  1. Line 138: What does AD stand for?

We are sorry to use the abbreviation here because we have mentioned the full name of AD — “atopic dermatitis” in the previous section (line 13 of the abstract).

  1. Line 154: A space is missing.

We apologize for our carelessness. We have added a space in line 170, the original sentence should be “and some kinds of cheeses contain large amounts of histamine”.

  1. Line 166: Please elaborate how certain foods can inhibit DAO.

We thank Reviewer#1 for raising this point, we have supplemented the mechanism of these certain foods inhibiting DAO in lines 178-181.

  1. Line 194: Awkward sentence.

We are sorry for our inaccurate English expression, and we have changed the sentence. We originally meant “Some patients develop wheal-like rash which looks like a symptom of urticaria”. We have modified the expression on lines 227-228.

  1. Figure 1: nasal mucosal is not a symptom.

We thank reviewer 1 for pointing out our negligence. It should be a problem with the typesetting of the picture, the respiratory symptoms in the picture should be "congestion of nasal mucosa".

  1. Line 234: I dont think this title reflects the contents of this section.

This section is mainly about those diseases that are easily confused with HIT. According to reviewer 1’s comment, we have changed the title to "Differential Diagnostic Exclusion of Other Diseases". (lines 274)

  1. Line 266: ELISA is not a histological analysis.

We thank reviewer 1 for pointing out that ELISA is indeed not a histological examination, and we have deleted the "histologically" in line 304.

  1. Lines 321-323: Do not use first person.

We apologize for our unprofessional expression. We have deleted the first person statement in " we use 1% (10 mg/ml) of histamine solution to pipet on the intact skin and then to prick the skin with lancets. We consider a wheal ≥ 3mm up to 50 minutes as positive. ", and have changed the expression to read "the intact skin was pipetted with 1% (10 mg/mL) histamine solution, and the skin was punctured with a lancet. A wheal ≥ 3 mm within 50 minutes was considered positive." (lines 366-368)

  1. Lines 327-329: This paragraph belongs to a separate section.

According to reviewer 1’s comment, we have divided this paragraph into a separate section.

  1. Lines 331-332: Awkward sentence.

We are very sorry for this confusing sentence. We have revised the expression of this sentence to "If patients have the corresponding clinical manifestations of HIT, doctors can treat them with a low histamine diet or supplementation with DAO." in lines 387-389.

  1. Lines 343-345: These are all routine laboratory techniques that do not sound expensive or inconvenient.

We thank reviewer 1 for raising this issue worth discussing. We thought that these methods are time-consuming in clinical practice for HIT patients. Serum histamine level is a kind of auxiliary test for diagnosis of HIT. Compared with its value, the cost might be relatively expensive.

  1. Lines 373-374: Grammatical error.

We would like to express our sincere appreciation for your meticulous review of our manuscript. We have changed “many studies have made specific recommendations for foods that people with histamine intolerance must be avoided” to “many studies have made specific recommendations about foods to avoid for people with HIT”. (lines 504-505)

  1. Line 397: Change ml to mL throughout the manuscript.

We are sorry for our mistake. We have changed the full text and Figure 5 from “ml” to “mL”.

Our revisions have been highlighted in yellow in the revised manuscript. Attached please find the revised version, which we would like to submit for your kind consideration.

Reviewer 2 Report

The authors reviewed histamine intolerance (HIF) as a kind of pseudoallergic reaction. The manuscript is well-written with two tables (foods rich in histamine and foods that cause HIF) and three figures (clinical manifestations of HIT, differential diagnosis of HIT, and treatment of severe urticaria caused by HIT).

However, it may be more helpful for general readers if the authors could schematically show causes of HIT (section 3.1) and diagnosis of HIT (section 6) as additional figures. Particularly, relationship between foods and other causes listed in the section 3.1 should be clearly shown with respect that HIT is food intolerance.

Author Response

Dear Prof. Naomi Chang (Editor), and Reviewers:

Thank you for your letter and for the reviewers’ comments concerning our manuscript entitled “Histamine intolerance—a kind of pseudoallergic reaction” (Manuscript ID: biomolecules-1607389). The comments are all valuable and were very helpful for revising and improving our manuscript. We have studied the comments carefully and hope that the corrections meet with your approval. The revised portions of the paper are highlighted in yellow. The main corrections in the paper and the responses to the reviewers’ comments are as follows:

To Reviewer 2

Comments to the Author
The authors reviewed histamine intolerance (HIF) as a kind of pseudoallergic reaction. The manuscript is well-written with two tables (foods rich in histamine and foods that cause HIF) and three figures (clinical manifestations of HIT, differential diagnosis of HIT, and treatment of severe urticaria caused by HIT).

However, it may be more helpful for general readers if the authors could schematically show causes of HIT (section 3.1) and diagnosis of HIT (section 6) as additional figures. Particularly, relationship between foods and other causes listed in the section 3.1 should be clearly shown with respect that HIT is food intolerance.

On behalf of my co-authors, we thank you very much for giving us an opportunity to revise our manuscript, we appreciate reviewer 2 for the positive and constructive comments and suggestions on our manuscript.

We have carefully examined reviewer 2‘s comments and have added a figure for section 3.1 (Figure 2, lines 196-200), and a table for section 6 (Table 3, line 486).

Additionally, we thank reviewer 2 for pointing out the inadequacies of our expression. We have made some additions in section 3.1. (lines 188-192)

Our revisions have been highlighted in yellow in the revised manuscript. Attached please find the revised version, which we would like to submit for your kind consideration.

Round 2

Reviewer 1 Report

The authors have addressed my comments and revised the manuscript accordingly.